# Unraveling Connective Tissue Growth Factor as a Therapeutic Target and Assessing Kahweol as a Potential Drug Candidate in Triple-Negative Breast Cancer Treatment

**DOI:** 10.3390/ijms242216307

**Published:** 2023-11-14

**Authors:** Jeong Hee Lee, Jongsu Kim, Hong Sook Kim, Young Jin Kang

**Affiliations:** 1Department of Biological Sciences, Sungkyunkwan University, Suwon 16419, Republic of Korea; jeonghee03@naver.com (J.H.L.); djk6293@gmail.com (J.K.); 2Department of Pharmacology, College of Medicine, Yeungnam University, Daegu 42415, Republic of Korea

**Keywords:** connective tissue growth factor (CTGF), kahweol, triple-negative breast cancer (TNBC)

## Abstract

Triple-negative breast cancer (TNBC) is characterized by aggressive behavior and limited treatment options, necessitating the identification of novel therapeutic targets. In this study, we investigated the clinical significance of connective tissue growth factor (CTGF) as a prognostic marker and explored the potential therapeutic effects of kahweol, a coffee diterpene molecule, in TNBC treatment. Initially, through a survival analysis on breast cancer patients from The Cancer Genome Atlas (TCGA) database, we found that CTGF exhibited significant prognostic effects exclusively in TNBC patients. To gain mechanistic insights, we performed the functional annotation and gene set enrichment analyses, revealing the involvement of CTGF in migratory pathways relevant to TNBC treatment. Subsequently, in vitro experiments using MDA-MB 231 cells, a representative TNBC cell line, demonstrated that recombinant CTGF (rCTGF) administration enhanced cell motility, whereas CTGF knockdown using CTGF siRNA resulted in reduced motility. Notably, rCTGF restored kahweol-reduced cell motility, providing compelling evidence for the role of CTGF in mediating kahweol’s effects. At the molecular level, kahweol downregulated the protein expression of CTGF as well as critical signaling molecules, such as p-ERK, p-P38, p-PI3K/AKT, and p-FAK, associated with cell motility. In summary, our findings propose CTGF as a potential prognostic marker for guiding TNBC treatment and suggest kahweol as a promising antitumor compound capable of regulating CTGF expression to suppress cell motility in TNBC. These insights hold promise for the development of targeted therapies and improved clinical outcomes for TNBC patients.

## 1. Introduction

Breast cancer is the most diagnosed invasive cancer and the leading cause of cancer death in women worldwide. The incidence of breast cancer in women has been gradually rising at a rate of approximately 0.5% per year since the mid-2000s [1,2]. Estrogen receptor (ER), progesterone receptor (PR), and human epidermal growth factor receptor 2 (HER2) assessments in breast cancer are prognostic and predictive markers for hormonal and anti-HER2-targeted therapy. Triple-negative breast cancer (TNBC), which accounts for 10% to 15% of breast cancers, has a negative expression of the three markers [3]. It also has malignant clinical phenotypes, such as visceral organs and brain metastases. Metastasis, which is a multiple process involving the local invasion and migration into adjacent tissues, intravasation, and proliferation, plays a pivotal role in contributing to the aggressiveness of cancer [4,5]. Compared to other subtypes of breast cancers, TNBC patients show a high recurrence rate within 5 years after surgery and a poorer prognosis, because there are no effective markers for TNBC-targeted therapies [6,7]. It also has been reported that TNBC patients are four times more likely to experience an internal metastasis than other types of cancer within five years of diagnosis [8]. To date, the discovery of novel therapeutic targets in TNBC treatment has not been sufficiently identified. Therefore, adjuvant studies of TNBC therapeutic targets are still essential for developing patient-specific treatment strategies and improving the prognosis of TNBC patients.

Connective tissue growth factor (CTGF), also known as *CCN2*, is a matricellular protein of the CCN family members (CCN1-6) and is well-known as a profibrotic agent. CTGF was reported to be overexpressed in various cancer cells such as breast, ovarian, skin, lung, and liver cancers, and it contributes to the malignancy of cancers [9,10,11,12,13]. CTGF overexpression is involved in migration, extracellular matrix (ECM) production, proliferation, chemotaxis, and drug resistance by which it contributes to the tumor’s development and progression [14]. Recently, increasing evidence indicates that CTGF plays a pivotal role in regulating cancer cell growth by being involved in the interaction between tumor cells and the tumor microenvironment and is related to a poor prognosis in most types of cancer [15]. Since the regulation of CTGF expression has a pivotal role in the pathophysiology of cancer, elucidating functions of CTGF as a prognostic molecule in cancer treatment may provide experimental evidence improving prognosis in various cancer types.

Kahweol, a coffee diterpene molecule, is extracted from arabica coffee beans and is mainly present as fatty esters. Recent studies have reported that kahweol has various biological and pharmacological activities such as antioxidant, anti-inflammatory, anti-diabetic, anti-angiogenic, antitumor, and apoptotic effects in many cell types [16,17]. So far, most kahweol studies have demonstrated its antitumor effects in several cancers. It has been reported that kahweol not only induces antiproliferative effect and apoptosis in several cancers including HER2-positive breast, colorectal, and non-small cell lung cancers [18,19,20] but also significantly inhibits cell migration in prostate and renal cancers [21,22]. In a previous study, we demonstrated that kahweol suppresses Angiotensin II-induced migration through the inhibition of CTGF expression in vascular smooth muscle cells (VSMCs) [23]. Few studies on the association between kahweol and CTGF have been additionally reported in the hepatic fibrosis [24,25]. Nevertheless, studies of CTGF and kahweol have not been demonstrated in cancer. In the present study, we reasoned that CTGF may function as a prognostic marker by involving the regulation of cancer signaling in TNBC. Thus, the potential function of CTGF as a prognostic marker and associations with kahweol are investigated.

## 2. Results

### 2.1. The Involvement of CTGF Expression in Survival and Cancer Progression in TNBC

The assessment of RNA expression levels was a starting point to determine the potential function of CTGF in TNBC. To investigate whether CTGF expression is involved in the survival of TNBC patients, survival and pathway analyses were performed. The RNA-Seq and clinical data were obtained from the broad global data assembly centered firehose (GDAC, https://gdac.broadinstitute.org/, accessed on 11 July 2021) and the subtype of all 1093 breast cancer patients was classified using the Prediction Analysis of Microarray 50 (PAM50) classification. The survival analysis was performed in the CTGF high-expression group vs. the CTGF low-expression group separately for all breast cancer (BRCA) patients, TNBC patients, and non-TNBC patients. Interestingly, the results revealed a significant difference in survival probability between the TNBC patients with a high CTGF expression and those with a low CTGF expression (Figure 1A). However, no such difference in survival was observed in the non-TNBC patients and total BRCA patient population (Appendix A). Patient characteristics in the high-CTGF group and low-CTGF group were analyzed, and no significant differences in age, living status, pathologic stage, and race and tumor, node, metastasis (TNM) stage were observed (Appendix A). The ER, PR, and HER2 status indicated TNBC characteristics (Appendix A). Next, to investigate the potential molecular mechanisms underlying CTGF’s role in TNBC, the pathway analysis was conducted. A total of 1358 genes with a *p*-value of <0.05 between the high- and low-CTGF expression TNBC groups were selected, and an over-representation analysis (ORA) was performed through ConsensusPathDB (CPDB, http://consensuspathdb.org/, accessed on 15 July 2021) [26]. We identified ten well-known cancer-related pathways including extracellular matrix organization, collagen formation, integrin, focal adhesion, ECM–receptor interactions, the inflammatory response pathway, vascular endothelial growth factor (VEGF)-VEGF receptor 2 (VEGFR2) signaling, phosphatidylinositol 3-kinase (PI3K)-protein kinase B (AKT) signaling, transforming growth factor-β (TGF-β) signaling and wingless-related integration site (Wnt) signaling. These pathways, which are involved in the development and progression of cancer, were significantly activated in the TNBC group with a high CTGF expression (Appendix A), suggesting an oncogenic role of CTGF in TNBC. Additionally, a gene ontology (GO) enrichment analysis was performed using Metascape and the database for annotation, visualization, and the integrated discovery (DAVID) database. The functional annotation analysis with Metascape found that the 1358 genes (*p*-value < 0.05) were significantly enriched in tumor survival including extracellular matrix organization, cell migration, cell adhesion, and the response to wounding (Figure 1B). A GO and Kyoto encyclopedia of genes and genomes (KEGGs) analysis based on DAVID was classified into biological process (BP), cellular components (CC), molecular function (MF), and KEGGs. The result from DAVID showed various cell migration- and proliferation-related pathways including extracellular matrix organization, focal adhesion, mitogen-activated protein kinase (MAPK), PI3K-AKT, and protein bindings such as integrin, collagen, and fibronectin (Figure 1C). Collectively, migration-related pathways including extracellular matrix organization, collagen formation, integrin, ECM–receptor interactions, and focal adhesion were prominent in CTGF-associated TNBC. Thus, we reconstructed a heatmap using the genes under the migration-related pathways from ORA (Figure 1D). Taken together, this suggests that CTGF expression in TNBC contributes to the cancer progression and clinical outcomes of TNBC.

### 2.2. CTGF Contributes to Cell Motility in TNBC

Since a negative correlation between CTGF expression and survival in TNBC patients was observed, the function of CTGF as a potential prognostic marker in TNBC treatment was determined. First, CTGF knockdown was employed by CTGF siRNA transfection in MDA-MB 231 cells, a TNBC cell line (Figure 2A). Then, we performed a transcriptomic analysis by RNA-Seq on control siRNAs or CTGF siRNAs. Whole transcriptomic profiles of CTGF siRNA-transfected cells were compared with the control siRNA-transfected group, and differentially expressed genes (DEGs) were identified by an application of the screening thresholds of |fold changes (FCs)| < 0.7 or >1.3 with a *p* value of <0.05. Among all the selected 1236 DEGs (9.6% of all genes), 381 genes (3% of all genes) were upregulated and 855 genes (6.6% of all genes) were downregulated in the siCTGF group compared to the siControl group in the MDA-MB 231 cells. The thirty most significantly upregulated and downregulated DEGs are shown in the Appendix A. To investigate the potential function of CTGF in TNBC, functional annotation and gene set enrichment analyses were performed. The functional annotation analysis with Metascape found that the DEGs between the siCTGF and siControl groups were significantly enriched in extracellular matrix organization, the actin filament-based process, chemotaxis, and the regulation of MAPK cascades (Figure 2B). The gene set enrichment analysis (GSEA) of the RNA-seq data shows significant enrichments for migration-related gene sets. The enrichments for downregulated gene sets were ‘ECM receptor interaction’, ‘cell adhesion molecules’, and ‘focal adhesion’. CTGF knockdown significantly reduced the enrichment scores of the three gene sets (Figure 2C). Furthermore, we investigated whether the results from the Metascape and GSEA by the RNA-seq data were consistent with the cell phenotype. Since metastasis is a multi-step process from cell migration to proliferation leading to cell motility [4,5], we investigated migratory phenotypes in vitro. The migratory and wound-healing capacity of the MDA-MB 231 cells were assessed using transwell and scratch assays. The depletion of CTGF led to a 77% decrease in cell migration when compared to the siControl group (Figure 2D). In parallel, in the control group, wound closure reached 85% while siCTGF achieved only 28% closure (Figure 2E). This observation indicates the significant involvement of CTGF in the motility of TNBC cells.

### 2.3. Kahweol Suppresses CTGF Expression and the Ability of Cell Migration in TNBC

In our previous study, kahweol suppressed CTGF expression in VSMCs [23]. To determine whether kahweol inhibits CTGF expression in TNBC, the mRNA level of CTGF was investigated. The MDA-MB 231 cells were treated with 50 µM of kahweol for 24 h. Kahweol significantly reduced the CTGF mRNA expression (Figure 3A). We performed a transcriptomic analysis by RNA-Seq on the control or kahweol treated group. Whole transcriptomic profiles of the kahweol-treated cells were compared with the control group. To investigate the potential function of kahweol in TNBC, a gene set enrichment of the kahweol-treated group compared to the control was implemented via GSEA using the RNA-seq data. GSEA showed significant enrichments for migration-related gene sets. The enrichments for the downregulated gene sets were ‘cell adhesion molecules’, ‘ECM receptor interaction’, ‘focal adhesion’, and ‘leukocyte transendothelial migration’. Kahweol significantly reduced the enrichment scores of the four gene sets (Figure 3B). To verify the specificity of the anti-migratory effect of kahweol, genes that remained unaffected by kahweol were explored. We identified 6174 genes (0.7 < FC < 1.3) when comparing the kahweol-treated cells to the untreated cells in the siControl-expressing cells. Similarly, in the siCTGF-expressing cells, we selected 6229 genes (0.7 < FC < 1.3) by comparing the kahweol-treated with the untreated cells. Particularly, cell cycle-related genes were not affected by kahweol in the presence or absence of CTGF (Appendix A), providing evidence that kahweol specifically influences the mobility function of MDA-MB 231 cells. Next, to confirm whether kahweol is involved in migratory phenotypes, consistent with the GSEA results, migratory ability was assessed in the MDA-MB 231 cells, and, consistent with the RNA-Seq data, kahweol attenuated migration by 78% compared to the control group (Figure 3C). These results suggest that kahweol reduces CTGF expression and has potential as an anti-migratory compound in TNBC treatment.

### 2.4. Kahweol Inhibits Cell Motility by Regulating CTGF Expression in TNBC

Next, we investigated whether kahweol inhibits cell motility via CTGF regulation, employing a recovery experiment strategy. As shown in Figure 3A, kahweol reduced the CTGF mRNA and protein levels by 0.6-fold and 0.3-fold, respectively, relative to the untreated control. However, rCTGF administration recovered CTGF expressions almost equivalent to the control sample. The effect of rCTGF was also examined (Figure 4A,B). Next, the anti-migratory effect of kahweol in the MDA-MB 231 cells was evaluated by a transwell migration assay. The rCTGF-administrated MDA-MB 231 cells showed an increased migration by 151% relative to the untreated control. Kahweol reduced migration levels by 22%, and rCTGF administration significantly reversed migration, increasing it by 92% (Figure 4C). Additionally, a wound-healing assay was performed for the same set of samples. Kahweol treatment recovered approximately 35% of the scratched area, and the rCTGF with kahweol significantly restored it to 40% in 24 h. The untreated control and rCTGF restored 50% and 70% of the scratched area in 24 h. In 48 h, three samples, the untreated control, rCTGF, and the rCTGF with kahweol showed an almost similar wound-closure area by restoring it by ~95%, but kahweol recovered only 68% of the scratched area (Figure 4D). Taken together, these results demonstrate that kahweol inhibits TNBC cell migration by regulating CTGF.

### 2.5. Kahweol Influences the Expression of Genes Associated with Migration by Modulating CTGF via ERK, P38, PI3K, AKT, and FAK Phosphorylation

To confirm the underlying mechanisms involved in the regulation of CTGF by kahweol, the expression levels of cell motility-related genes including vinculin (VCL), collagen type IV alpha 4 chain (COL4A4), collagen type XI alpha 1 chain (COL11A1), and matrix metallopeptidase 1 (MMP1) were investigated. These genes were involved in migration-related pathways based on the transcriptomic analysis of TCGA TNBC patients, as shown in Figure 1. The relative expression levels of VCL, COL4A4, COL11A1, and MMP1 were significantly lower in the CTGF low-expression patients compared to the high-CTGF-expression group (Figure 5A and Appendix A). A further investigation was conducted to analyze the mRNA expression levels of these genes in the MDA-MB 231 cells following siCTGF transfection and rCTGF administration. In line with the patient data depicted in Figure 5A, siCTGF transfection resulted in a reduction in the expression of migration-related genes, whereas rCTGF caused an increase in their expression (Figure 5B,C). Kahweol treatment led to a suppression of the gene expression in comparison to the untreated control, and rCTGF administration significantly reversed this inhibitory effect (Figure 5C). Additionally, the underlying molecular mechanisms involved in CTGF were investigated by examining the expression and activation of MAPKs, PI3K/AKT, and focal adhesion kinase (FAK) proteins. These proteins have established their roles in cell migration [27,28,29]. The phosphorylation of extracellular signal-regulated kinase (ERK), P38, PI3K, AKT, and FAK was diminished by siCTGF, while the phosphorylation of c-Jun N-terminal kinase (JNK) was increased (Figure 5D). Furthermore, whether these signaling proteins are involved in the regulation of CTGF by kahweol was investigated. Reductions in the ERK, P38, PI3K, AKT, and FAK phosphorylation by kahweol were notably reversed by the administration of rCTGF (Figure 5E). Taken together, kahweol influences the expression of migration-related genes by regulating CTGF through the phosphorylation of ERK, P38, PI3K, AKT, and FAK. These findings suggest that kahweol has the potential to serve as a CTGF-regulating agent in the treatment of TNBC.

## 3. Discussion

Breast cancer is the most diagnosed invasive cancer and the leading cause of cancer death in women. It has four primary molecular subtypes, defined in large part by hormone receptors, ER and PR, and HER2 receptors. TNBC subtype is defined by the absence of ER and PR expressions as well as the absence of the HER2 receptor expression. It is characterized by its biological aggressiveness, worse prognosis, and lack of therapeutic targets in contrast with non-TNBC. TNBC is not sensitive to endocrine therapy or HER2 treatment, and standardized TNBC-treatment regimens are still lacking [3,6]. Based on the clinical challenge in treating TNBC, we have identified that kahweol, a naturally occurring compound, shows promise as a potential treatment for TNBC by targeting CTGF. Several studies have demonstrated oncogenic functions for CTGF in various types of cancers including melanoma, ovarian cancer, prostate cancer, colorectal cancer, lung adenocarcinoma, hepatocellular carcinoma, and breast cancer [10,12,13,30,31,32,33]. Although a previous study demonstrated the oncogenic role of CTGF in breast cancer, it had limitations concerning clinical implications [10]. In this previous study, the authors manipulated the CTGF-expression level and observed its tumorigenic effects in MCF-7 cells and MDA-MB 231 cells. In the present research, we aimed to explore the prognostic significance of CTGF and to discover therapeutical approaches within the realm of clinical practice.

Metastasis, which is an important risk factor in determining survival, is fundamentally derived from cell migration [4,5]. TNBC has shown a significant prevalence of brain, lung, and distant nodal metastases. The incidence of brain metastasis in TNBC is approximately 25–46%, which is a 2–5 times higher risk of developing brain metastases compared to Luminal A (8–15%) and B (11%) [34,35]. According to a recent meta-analysis, roughly 33% of individuals with TNBC will ultimately experience brain metastasis [36]. In a previous study, recombinant CTGF proteins increased VSMC migration and the expression of FAK and the yes-associated protein (YAP), which is related to the formation of focal adhesion and cell migration [23]. It has been demonstrated that the upregulation of CTGF leads to an increase in the expression of matrix metalloproteinases, such as matrix metalloproteinase (MMP)-2 and MMP-3, consequently stimulating cell migration in osteosarcoma [31]. It has also been reported that CTGF’s binding to α6β1 mediates collagen deposition [37]. In line with this, we further investigated CTGF-associated genes and signaling pathways in TNBC. Consistent with previous reports, CTGF plays a functional role in signaling pathways associated with migration in TNBC patients. (Figure 1). Additionally, it is affirmed in MDA-MB 231 cells by CTGF knockdown (Figure 2). These studies confirm that since CTGF plays a pivotal role in TNBC exacerbation via its involvement in cell motility, the improvement of TNBC prognosis could be expected by regulating the CTGF level. They also provide evidence that CTGF is a potential therapeutic target and prognostic marker in TNBC.

In this study, we have identified kahweol as a potential drug candidate targeting CTGF in TNBC. Previous research has reported that kahweol acetate inhibits migration and proliferation in prostate cancer cells [21]. Additionally, recent updates on the functional impacts of kahweol and cafestol on cancers have shown that kahweol exerts antitumor effects on breast cancer cell lines [38]. Consistent with these findings, another study showed that kahweol treatment in MDA-MB 231 resulted in the inhibition of cell proliferation along with the induction of apoptosis by the increased production of reactive oxygen species and cytotoxicity [39]. Considering the role of CTGF in cell proliferation and migration in cancer (Figure 1 and Figure 2 and Appendix A), we further investigated the potential anti-cancer effects of kahweol through CTGF inhibition. In this study, we confirmed that kahweol inhibited CTGF expressions in MDA-MB 231 cells. Furthermore, our functional annotation results show that kahweol was involved in motility, which was confirmed to affect migration through in vitro transwell experiments (Figure 3 and Figure 4). In Figure 5, by bridging basic research with the clinical implications of kahweol and CTGF, our results carry a more substantial impact. We observed that kahweol reduced the expression of CTGF-associated migration genes. Kahweol led to the decreased phosphorylation of ERK, P38, PI3K/AKT, and FAK. Notably, this effect was restored when rCTGF was administered, indicating that kahweol impacts the expression of genes associated with migration by modulating CTGF via the ERK, P38, PI3K, AKT, and FAK phosphorylation. Interestingly, we observed an increase in JNK phosphorylation upon kahweol treatment. It is worth noting that previous studies showed that MAPK proteins including ERK, P38 and JNK did not consistently exhibit concurrent effects, suggesting the possibility of unconfirmed impacts in specific contexts [40,41]. It has been demonstrated that kahweol significantly suppresses MMP-9 expressions by inhibiting the MAPK phosphorylation and nuclear factor kappa B (NF-κB) signaling pathways, effectively attenuating metastasis in human fibrosarcoma cells [42]. Moreover, kahweol plays its anti-inflammatory and anti-atherogenic effects by blocking the activation of the Janus kinase 2 (JAK2)-PI3K/Akt-NF-κB pathway induced by tumor necrosis factor α (TNFα). It also downregulates pathways that influence the expression and interaction of cell adhesion molecules in endothelial cells [43]. CTGF enhances cell–cell communication in chondrocytes through the PI3K/Akt signaling pathway [44]. FAK phosphorylation leads to the recruitment and activation of focal adhesion components such as vinculin, paxillin, and Src in response to cell–ECM adhesion through integrins [45]. These studies further support the involvement of ERK, P38, PI3K/AKT, and FAK in CTGF regulation by kahweol in TNBC. Nevertheless, in terms of underlying mechanisms, our study of the mechanisms associated with cancer metastasis and the therapeutic implications of kahweol in TNBC still remains in its infancy. Previous studies have shown YAP, a well-known transcription regulator of CTGF, to promote focal adhesion by activating FAK in breast cancer and to induce CTGF expression via ERK in hepatocytes [46,47]. In our previous study, we observed that kahweol reduced CTGF expression and migration in VSMCs while suppressing the protein expression of FAK, YAP, ERK1/2, fibronectin, and collagen III [23]. These studies provide valid evidence that YAP, as a starting point of our future study, acts as a potential transcriptional mediator of CTGF regulatory mechanisms in the context of therapeutic TNBC studies. Overall, the therapeutic significance of CTGF and kahweol, as revealed through our study, has the potential to contribute to the prognosis and treatment of TNBC. To develop kahweol as an anti-cancer agent in clinics, more comprehensive clinical investigations are called for. Consequently, the therapeutic function of kahweol in conjunction with their mechanism of action and pharmacokinetic profile, such as absorption, bioavailability, metabolism, and elimination, should be evaluated. Further explorations into the underlying molecular mechanisms could open up avenues for TNBC-treatment research and offer promising therapeutic approaches.

## 4. Materials and Methods

### 4.1. Reagents and Antibodies

The MDA-MB 231 cell line was bought from ATCC (Manassas, VA, USA). The recombinant human CTGF was purchased from PEPROTECH (East Windsor, NJ, USA). Kahweol was purchased from Santa Cruz Biotechnology (Dallas, TX, USA). Antibodies were purchased from the following vendors: CTGF (Dallas, TX, USA) and β-actin from Sigma-Aldrich (St. Louis, MO, USA), ERK, JNK, phospho-ERK, and phospho-JNK from Santa Cruz Biotechnology (Dallas, TX, USA) and P38, PI3K, AKT, FAK, phospho-P38, phospho-PI3K, phospho-AKT, and phospho-FAK from ABclonal (Woburn, MA, USA). Dulbecco’s modified Eagle’s medium (DMEM), the fetal bovine serum (FBS), penicillin/streptomycin antibiotics, and trypsin-EDTA solution were purchased from GenDEPOT (Katy, TX, USA).

### 4.2. Cell Culture and Kahweol Preparation

MDA-MB-231 human breast cancer cell lines were maintained in DMEM supplemented with 10% FBS and 1% antibiotics (penicillin, 10,000 U/mL; streptomycin, 10,000 µg/mL) at 37 °C in a humidified atmosphere with 5% CO_2_. MDA-MB 231 cells were prepared at 70–90% confluence for the experiments.

The solubility of kahweol in the product’s datasheet (Santa Cruz, Cat. sc-203089) is described as soluble in a 1:50 solution of ethanol:PBS (pH 7.2) (~0.02 mg/mL), DMF (~5 mg/mL), DMSO (~3 mg/mL), and ethanol (~5 mg/mL). In our experiments, we reconstituted the powdered kahweol into a solution of 10 mM of kahweol by adding DMSO.

### 4.3. Total RNA Sequencing

Total RNA was isolated using the Apure Total RNA kit (AP BIOTECH, Buenos Aires, Argentina). RNA sequencing was performed at MacroGen Inc. (www.macrogen.com, accessed on 2 July 2021, Seoul, Republic of Korea). The RNA concentration was calculated by Quant-IT RiboGreen (Invitrogen, Waltham, MA, USA, #R11490). To assess the integrity of the total RNA, samples were run on the TapeStation RNA screentape (Agilent, Santa Clara, CA, USA, #5067-5576). Only high-quality RNA preparations, with RINs greater than 7.0, were used for the RNA library’s construction.

A library was independently prepared with 1 ug of total RNA for each sample by the Illumina TruSeq Stranded Total RNA Sample Prep Kit (Illumina, Inc., San Diego, CA, USA). The rRNA in the total RNA was depleted by the Ribo-Zero kit. After the rRNA was depleted, the remaining RNA was purified, fragmented, and primed for cDNA synthesis. The cleaved RNA fragments were copied into first-strand cDNA using reverse transcriptase and random hexamers. This was followed by second-strand cDNA synthesis using DNA Polymerase I, RNase H, and dUTP. These cDNA fragments then went through an end repair process, the addition of a single ‘A’ base, and then the ligation of the adapters. The products were then purified and enriched with PCR to create the final cDNA library. The libraries were quantified using qPCR, according to the qPCR Quantification Protocol Guide (KAPA Library Quantification kits for Illumina Sequencing platforms), and were qualified using the TapeStation D1000 ScreenTape (Agilent Technologies, Waldbronn, Germany). Indexed libraries were then submitted to an Illumina NovaSeq6000 (Illumina, Inc., San Diego, CA, USA), and the paired-end (2 × 100 bp) sequencing was performed by Macrogen Incorporated.

### 4.4. Bioinformatics Analysis

Gene ontology (GO) and the Kyoto Encyclopedia of Genes and Genomes (KEGGs) pathway enrichment analyses were performed using the online tools DAVID and Metascape. The GO functional enrichment analysis, which included molecular function (MF), biological process (BP), and cellular component (CC) analyses, and the KEGGs pathway analysis of the DEGs were performed using the R package clusterProfiler. Gene set enrichment analysis (GSEA) v7.0 was used to explore the pathways and gene sets associated with CTGF and kahweol in MDA-MB 231 cells. The *p*-value < 0.05 was utilized to define the statistical significance of GSEA throughout this study.

### 4.5. TCGA Data Acquisition and Gene Selection

The Cancer Genome Atlas Breast Invasive Carcinoma (TCGA BRCA) mRNA expression data were acquired from GDAC’s site. Clinical data including the survival status of TCGA BRCA patients and subtypes depending on PAM50 criteria were obtained from The Cancer Genome Atlas Network [48]. The TCGA BRCA TNBC patient group (n = 98) and the non-TNBC patient group including Luminal A, Luminal B, HER2, and Normal-like (n = 424) were classified by PAM50 criteria. One patient in the TNBC group and nine patients in the non-TNBC group were excluded due to insufficient clinical information, resulting in a final analysis with the TNBC group (n = 97) and the non-TNBC group (n = 415). The Pearson correlation test was performed to identify correlations between the mRNA expression level of CTGF and mRNA expression levels of other genes in the TNBC group. Genes were selected by a *p*-value of < 0.05 between the high- and low-CTGF-expression TNBC groups.

### 4.6. Pathway Analysis and Clinical Data Visualization

A total of 1358 genes were included in the over-representation analysis (ORA) via ConsensusPathDB (CPDB, http://cpdb.molgen.mpg.de/, accessed on 20 July 2021). Pathways were selected with these criteria: a minimum overlapping input list (n = 2) and a *p*-value cutoff (*p*-value < 0.01). Pathway databases (Wikipathways, SMPDB, KEGG, Reactome, PharmGK, PID, Biocarta, Ehmn, Humancyc, INOH, Netpath, and Signalink) were used [49,50,51,52,53,54,55,56,57,58,59].

The analysis in this study was conducted using the R software version 3.6.3. The Complexheatmap package was employed to visualize mRNA expression levels. Survminer was used for survival analysis. For visualization, GGally and ggplot2 were utilized. Individual gene expressions were visualized by box plots using ggplot2, and statical analyses were performed using R. To visualize the migration-related pathways, a flow diagram was created using the SankeyMATIC online tool (https://sankeymatic.com/, accessed on 20 October 2021). To determine the cutoff for the CTGF high expression, a threshold of CTGF-expression levels (RPKM) > 3300 was selected, resulting in the most significant survival and *p*-value between the CTGF high-expression group and the low-expression group. For the network analysis and visualization, Cytoscape version 3.9.0 was employed [60].

### 4.7. Knockdown of CTGF Expression Using Small Interfering RNA (siRNA)

For the knockdown of CTGF expression, the following siRNA against CTGF mRNA was synthesized: 5′-GCACCAGCAUGAAGACAUA-3′ (Genolution Inc., Seoul, Republic of Korea). The negative control siRNA (Bionics, Seoul, Republic of Korea) was purchased for use in the control experiments. MDA-MB 231 cells were grown to 60% confluence in 60 mm culture dishes and were transfected with 100 nM of CTGF siRNA or the negative control siRNA using DharmaFECT 2 (Dharmacon Research Inc., Lafayette, CO, USA). After incubation for 6 h at 37 °C, the DharmaFECT mixtures were washed out and cells were incubated in a serum-free medium for 24 h before kahweol treatment.

### 4.8. Western Blot Analysis

MDA-MB 231 cells were prepared in a serum-free medium. They were treated with rCTGF (400 ng/mL) for 55 min, followed by the addition of kahweol (50 µM) for 5 min. After that, cells were lysed with Pro-Prep (iNtRON, Seongnam, Republic of Korea), supplemented with a 10 µM protease inhibitor cocktail and a 10 µM phosphatase inhibitor cocktail, and lysates were incubated on ice for 30 min and centrifuged at 15,000× *g* for 10 min at 4 °C. Protein concentrations were determined using a protein assay reagent from Bio-Rad (Hercules, CA, USA). Briefly, equal amounts of protein were mixed with a 5× SDS sample buffer and heated for 5 min at 95 °C before loading. Equal quantities of the protein (20 µg) were separated via sodium dodecyl sulfate-polyacrylamide gel electrophoresis (SDS-PAGE) for 120 min at 60–90 V then transferred onto a nitrocellulose membrane (GE Healthcare Life Sciences, Pittsburgh, PA, USA) for one hour at 100 V, and membranes were blocked with 5% non-fat milk in TBS containing 0.05% Tween 20 (TBS-T) for one hour at room temperature and then incubated overnight with primary antibodies (1:1000). Membranes were then washed in TBS-T and incubated for one hour at room temperature in 5% non-fat milk containing anti-rabbit or anti-mouse IgG. After washing with TBS-T, signals were visualized by using electrochemiluminescence (ECL)-detection reagents (Millipore, CA, USA), according to the manufacturer’s instructions.

### 4.9. Quantitative Real-Time RT-PCR (qRT-PCR)

MDA-MB 231 cells were prepared in a serum-free medium. They were treated with rCTGF (400 ng/mL) for 1 h, followed by the addition of kahweol (50 uM) for 24 h. After that, the total RNA was isolated using the APure total RNA kit (APBIO, Namyangju, Republic of Korea), and a reverse transcription (RT) reaction was conducted using the amfiRivert Reverse Transcriptase kit (GenDEPOT, TX, USA), following the manufacturer’s instructions. qRT-PCR was conducted with 1 µL of template cDNA and power SYBR Green (GenDEPOT, TX, USA), using the QuantStudio1 Real-Time PCR system (Thermo, Waltham, MA, USA). Quantification was performed by the efficiency-corrected ∆∆Cq method. The primers used to amplify the DNA sequences were as follows: CTGF (forward: 5′-ACC TGT GGG ATG GGC ATC T-3′; reverse: 5′-CAG GCG GCT CTG CTT CTC TA-3′); VCL (forward: 5′-CCA AAA CAT GTC TCC TAT ATC CTG G-3′; reverse: 5′-GAA GTG TCC TTC AGA CAG GG-3′); COL4A4 (forward: 5′-TGA AGG GAA ATC CCG GTG TG-3′; reverse: 5′-CAG GTG GCT CTA CCA ACA GG-3′); COL11A1 (forward: 5′-TGG TGA TCA GAA TCA GAA GTT CG-3′; reverse: 5′-AGG AGA GTT GAG AAT TGG GAA TC-3′); and GAPDH (forward: 5′-GGA TTT GGT CGT ATT GGG-3′; reverse: 5′-GGA AGA TGG TGA TGG GAT T-3′). The PCR conditions were as follows: initial denaturation at 95 °C for 10 min, 95 °C for 10 s, 60 °C for 15 s, and 72 °C for 1 min.

### 4.10. Wound Scratch Assay and Transwell Chamber Assay

Cells were grown to 90% confluence in 12-well plates in DMEM supplemented with 10% FBS, and the medium was replaced with 5% FBS-supplemented DMEM. Wounds were made with a sterile 200 µL pipet tip by drawing a line through the plated cells perpendicular to the line above. Cells were siCTGF transfected or pretreated with rCTGF (400 ng/mL) for 1 h, and then kahweol (50 μM) was added to the medium in the presence of rCTGF and incubated for 48 h. Images were acquired by using a phase-contrast microscope (4×). To perform the transwell assay, cells were seeded in the inner chamber of the transwell plate and treated under the same conditions as described for the wound-healing assay. For the transwell assay with the siCTGF condition, transfected cells were harvested and seeded in the inner chamber. At the assay’s conclusion, cells were fixed by 4% paraformaldehyde and visualized by crystal violet staining. Non-migrated cells were scraped off with cotton swabs, and then migrated cells were photographed under a light microscope (40×) (OLYMPUS, Tokyo, Japan). The number of migrated cells were quantified using image J (https://imagej.net/ij/).

### 4.11. Statistical Analysis

Excel (Microsoft 365 enterprise, Redmond, WA, USA) was used for data acquisition and analysis. Differences between data sets were assessed by Bonferroni’s *t*-test. Results are presented as the means ± standard deviation (SD) from at least three independent experiments, and *p*-values of < 0.05 were considered significant.

## 5. Conclusions

In conclusion, our current study identifies CTGF as a promising prognostic marker and therapeutic target, offering valuable insights into predicting the prognosis of TNBC treatment. Additionally, we discovered that kahweol, a potential antitumor compound, effectively regulates key signaling pathways such as ERK, P38, PI3K/AKT, and FAK which are involved in CTGF regulation. These findings offer valuable insights for further therapeutic studies targeting TNBC and hold promise for the development of novel treatment approaches, ultimately contributing to improved patient outcomes in the battle against this aggressive form of breast cancer.

## Figures and Tables

**Figure 1 ijms-24-16307-f001:**
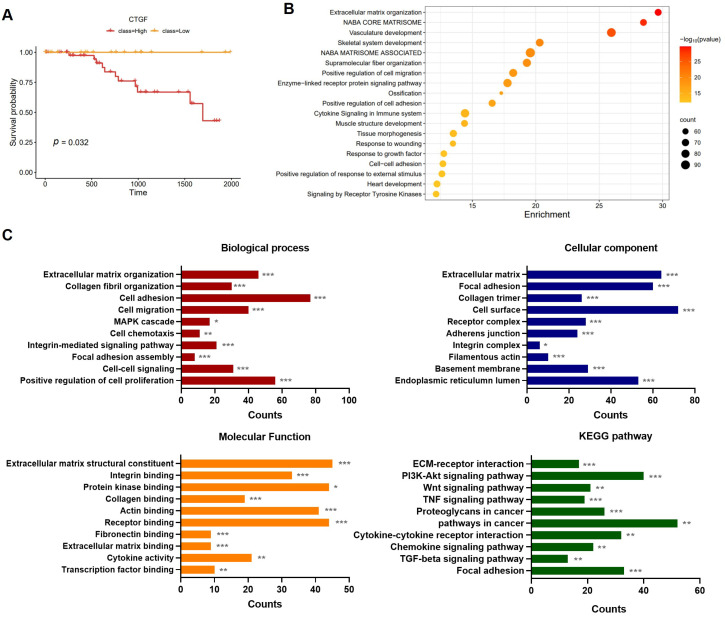
CTGF implication in the survival and cancer progression of TNBC. (**A**) Kaplan–Meier survival plot for TCGA TNBC patient showing overall survival in the CTGF high-expression group (red, n = 65) and CTGF low-expression group (yellow, n = 32). (**B**) An enrichment bubble plot visualizing the selected genes. Pathway enrichment of differentially expressed genes (DEGs) between the high-CTGF and low-CTGF groups is presented; darker colors indicate lower *p*-values. (**C**) Results of GO enrichment analysis and KEGG pathway analysis for DEGs (*p*-value < 0.05) in the high-CTGF and low-CTGF groups. (**D**) Sankey plot with selected pathways via CPDB (*p*-value < 0.05 in over-representation analysis) based on mRNA expressions in TNBC patients (n = 97). Five migration-related pathways and genes are indicated in the Sankey plot. * *p* < 0.05, ** *p* < 0.01, *** *p* < 0.005.

**Figure 2 ijms-24-16307-f002:**
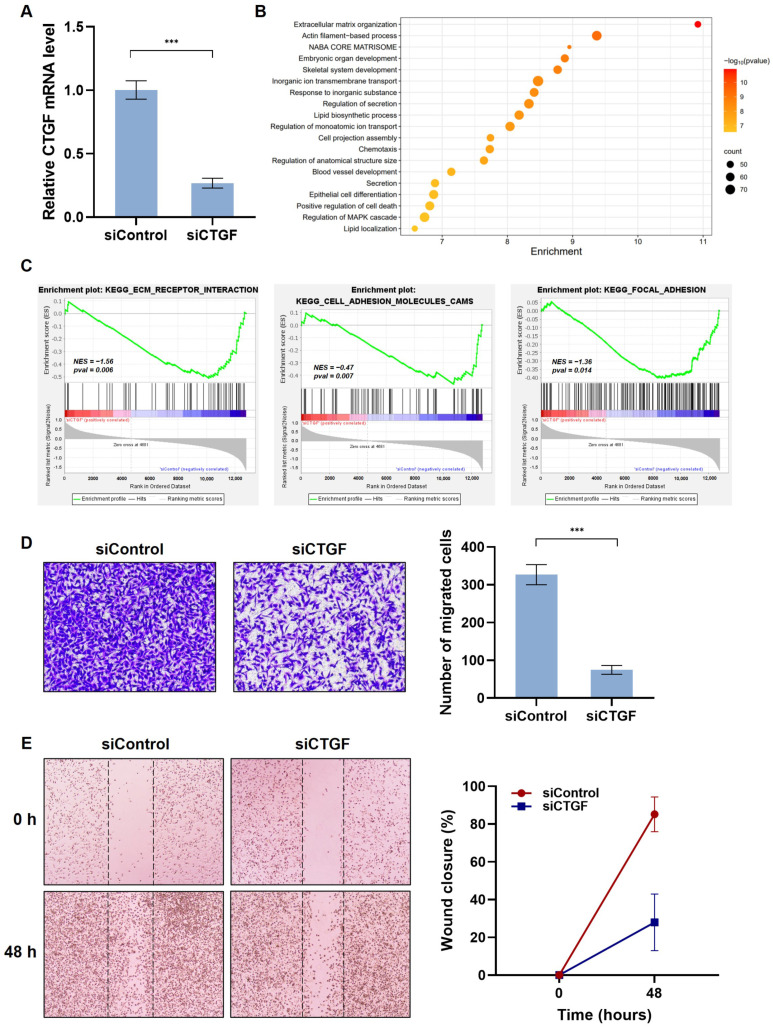
CTGF involvement in TNBC cell motility. MDA-MB 231 cells were transfected with 100 nM siRNA specific for the CTGF gene or control siRNA. (**A**) The mRNA level of CTGF was determined by qRT-PCR analysis. (**B**) Enrichment bubble plot visualizing the selected genes. Pathway enrichment of DEGs between control siRNA and CTGF siRNA groups is presented; darker colors indicate lower *p*-values. (**C**) Three significant enrichment plots of pathway enrichment analysis of DEGs using GSEA. GSEA-based KEGG analysis-enrichment plots of representative gene sets: ECM–receptor interactions, cell adhesion molecules, and focal adhesion. The green line represents the enrichment profile. (**D**) For selective migration assay, a transwell assay was performed. The negative control siRNA or CTGF siRNA-transfected cells were seeded on the inner chamber. After fixing, cells were visualized by crystal violet staining. Unmigrated cells were scraped off, and the migrated cells were counted under a light microscope. (**E**) Cells were scratched with a micropipette tip to form a cell-free (wounded) area and were transfected with siCTGF. Wound areas are visualized using a phase-contrast microscope. The distance migrated was determined as the average of the distances of each cell from the wound boundary. The results are representative of three independent experiments. The data are expressed as means ± SD. *** *p* < 0.005.

**Figure 3 ijms-24-16307-f003:**
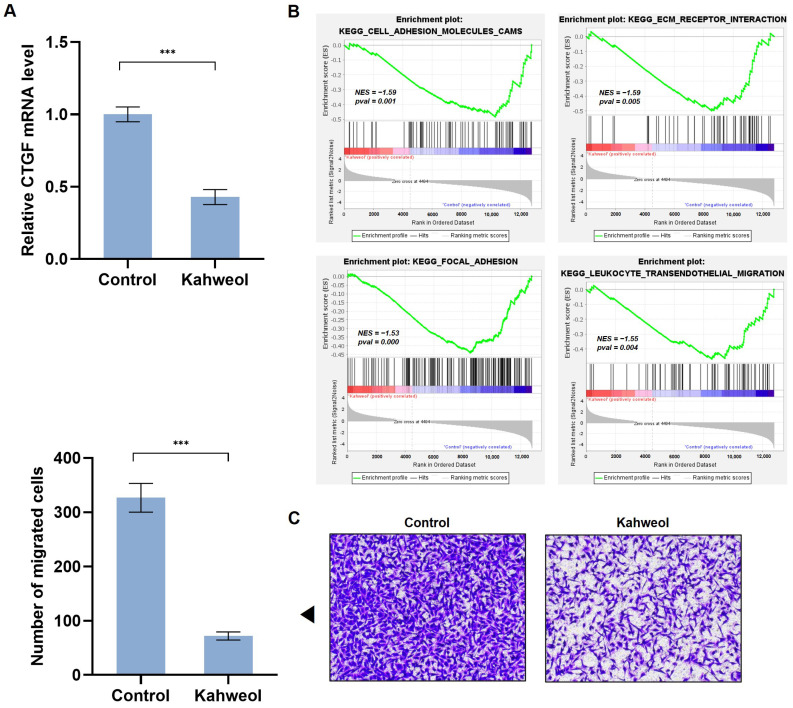
Anti-migratory effect of kahweol in MDA-MB 231 cells. MDA-MB 231 cells were treated with kahweol (50 µM) for 48 h. (**A**) mRNA level of CTGF was determined by qRT-PCR analysis. (**B**) Four significant enrichment plots of the pathway enrichment analysis of DEGs using GSEA. GSEA-based KEGG analysis-enrichment plots of representative gene sets: cell adhesion molecules, ECM–receptor interactions, focal adhesion, and leukocyte transendothelial migration. The green line represents the enrichment profile. (**C**) For the selective migration assay, a transwell assay was performed. Kahweol-treated cells were seeded on the inner chamber for 48 h. After fixing, cells were visualized by crystal violet staining. Unmigrated cells were scraped off, and the migrated cells were counted under a light microscope. The arrow indicates statistical graphs illustrating the results of the transwell analysis. Results are representative of three independent experiments. Data are expressed as means ± SD. *** *p* < 0.005.

**Figure 4 ijms-24-16307-f004:**
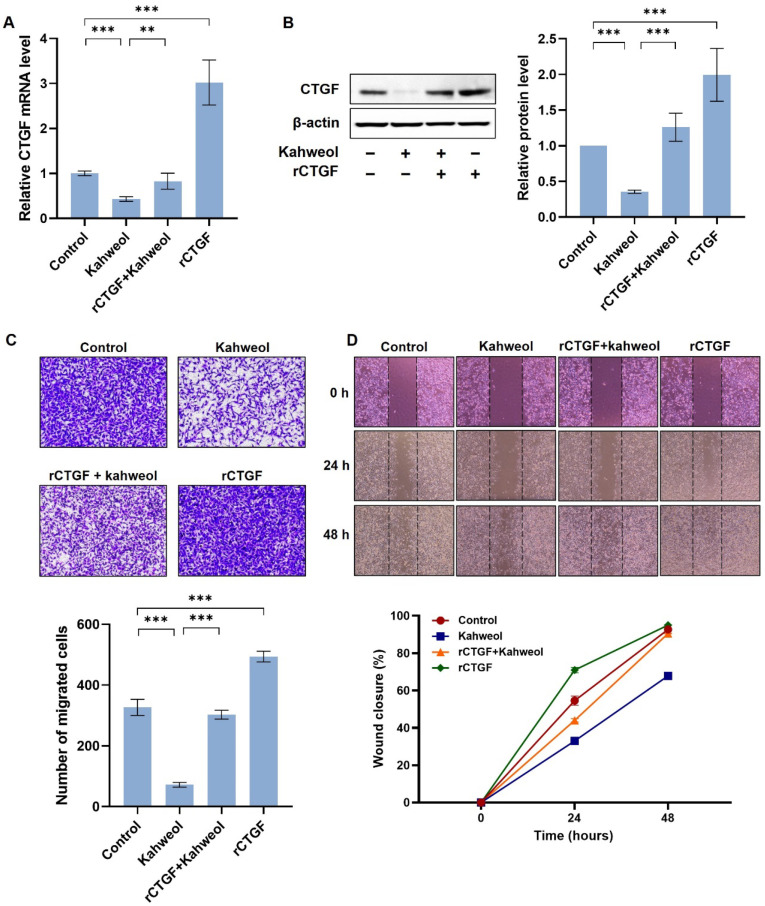
Kahweol’s effect on cell motility in MDA-MB 231 cells. MDA-MB 231 cells were pretreated with rCTGF (400 ng/mL) and kahweol (50 µM) at one-hour intervals for 24 h. (**A**) mRNA level of CTGF was determined by qRT-PCR analysis. (**B**) Western blot analysis of CTGF in rCTGF and kahweol-treated MDA-MAB 231 cells. (**C**) For the selective migration assay, a transwell assay was performed. siRNA-transfected cells were treated with kahweol and seeded on the inner chamber for 48 h. After fixing, cells were visualized by crystal violet staining. Unmigrated cells were scraped off, and the migrated cells were counted under a light microscope. (**D**) Cells were scratched with a micropipette tip to form a cell-free (wounded) area and were pretreated with rCTGF and kahweol at one-hour intervals and then stimulated. Wound areas are visualized using a phase-contrast microscope. The distance migrated was determined as the average of the distances of each cell from the wound boundary. Results are representative of three independent experiments. Data are expressed as means ± SD. ** *p* < 0.01, *** *p* < 0.005.

**Figure 5 ijms-24-16307-f005:**
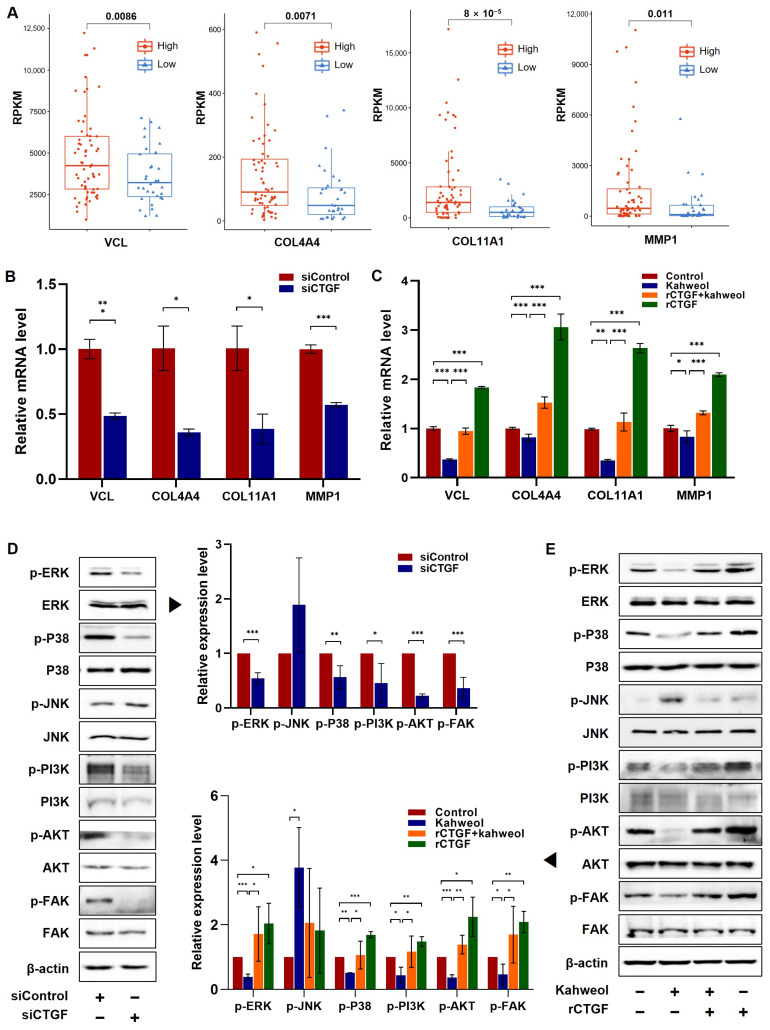
The involvement of MAPKs, PI3K/AKT, and FAK in the mechanisms underlying the kahweol effect on MDA-MB 231 cells’ motility. (**A**) The genes were selected from migration-related pathways in the RNA-seq results of clinical patient data and were plotted. (**B**,**D**) Cells were transfected with 100 nM siRNA specific for the CTGF gene or control siRNA. (**C**,**E**) Cells were pretreated with rCTGF (400 ng/mL) and kahweol (50 µM) at one-hour intervals for 24 h. (**B**,**C**) The mRNA levels of VCL, COL4A4, COL11A1, and MMP1 were determined by qRT-PCR analysis. Results are representative of three independent experiments. (**D**,**E**) Protein levels of phosphorylated MAPKs, PI3K/AKT, and FAK were determined by Western blotting with specific antibodies. The arrow indicates statistical graphs illustrating the results of the Western blot analysis, respectively. Bar graphs present the densitometric quantification of the Western blot bands. Data are expressed as means ± SD. * *p* < 0.05, ** *p* < 0.01, *** *p* < 0.005.

## Data Availability

The datasets generated during and/or analyzed during the current study are available from the corresponding author on reasonable request.

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
