# Peer review of "Unraveling Connective Tissue Growth Factor as a Therapeutic Target and Assessing Kahweol as a Potential Drug Candidate in Triple-Negative Breast Cancer Treatment"

_ijms, 2023, doi:10.3390/ijms242216307_

Round 1

Reviewer 1 Report

Comments and Suggestions for Authors

In this manuscript, the Authors show how the reduction of CTGF level interferes with different processes involved in TNBC progression, especially in migration, invasion, and metastases formation. Furthermore, they demonstrate that Kahweol, a coffee diterpene molecule, controls TNBC progression by lowering the CTGF mRNA level.

They assert that the low CTGF level interferes with the activity of different effectors such as ERK, p38, Akt, PI3K, and others.

The manuscript possesses some inaccuracies.

The authors should compare the CTGF level between normal and different cancer cell lines.

The authors should show in the experiments represented in Figs 4 and 5, the effect of CTGF siRNA too.

In Fig 5, panel C, they should explain why not all the effectors show a similar activation/inactivation between the treatments (the differences between untreated and Kahweol are not the same as those between cCTGF and cCTGF plus Kahweol. Could the Kahweol work in a dose-dependent way? Why the PI3K and AKT activations do not match? pJNK is increased by Kahweol treatment and only pFAK, PI3K, and slightly pP38 and pERK seem to be controlled by Kahweol.

Another question is: By which receptor kahweol could act?

Please check if all the abbreviations have the complete name the first time they are used.

Author Response

We sincerely thanks for the reviewer's careful readings to review the submitted article. The reviewers' comments on revisions were very helpful in improving the quality of the submitted paper. We are encouraged to revise extensively and endeavor to make better manuscript according to reviewer’s comment. The changes in manuscript are identified by page and paragraph and noted by the red color. We really expect the reviewer's positive consideration. 

The response report is attached as a word file.

Reviewer 2 Report

Comments and Suggestions for Authors

In this report, the authors investigate the role of CTGF and kahweol in migration of invasive breast cancer. They mine some databases to identify migration markers and to bench experiments to test the role of CTGF and kahweol. The premise is valid, but some experiments and results require re-assessment or more explanation before publication. Please see below my comments.

Major revisions

1.       Kahweol inhibits the expression of CTGF as well as other genes shown in figure 5. The authors conclude that Kahweol regulates motility through downregulation of CTGF and these genes. It is clear that motility is regulated when this chemical is used, but I am not convinced that it is due to changes in expression of these genes. There is no negative control. What if the chemical inhibits global gene expression? Can the authors find a gene or set of genes preferably in a non-motility related pathway that this chemical does not change expression?

2.       The statement at lines 252 and 253 does not quite represent what the figure is showing. First, not all protein levels equally restored by addition of rCTGF. Compare, for example, AKT and ERK. Second, in most cases is phosphorylation status and not expression that is affected. Take for example ERK. You can clearly see that the expression levels are equal in all lanes, but phosphorylation status is changed. Finally, what do the authors mean by “significantly”. Have statistics been done? Otherwise is just a visual inspection and the word “significantly” is not an accurate scientific conclusion. Adding to the confusion is that the data in B appears to represent the quantification of C from the figure legend. Is that true? IF that is so, please make it clear. Additionally, please indicate how many times the experiment was done and include error bars.

Minor revisions

1.       Please use the same type of reference. Compare line 71 with line 74.

1.       Please be consistent with gene nomenclature, either italicized or not italicized. Check for example line 83 on CTGF.

2.       Table 1 should be relegated to supplemental as it does not show significant p-values.

3.       Figure 1B is impossible to read. I suggest that this part B be made a stand-alone figure and the panels be blown up on an entire page. As it stands, it is unreadable. Either that or relegate that panel to supplemental and keep D only in the figure.

Comments on the Quality of English Language

Some improvements are needed.

Author Response

(The authors gave the same response as above.)

Reviewer 3 Report

Comments and Suggestions for Authors

In this study the authors, attempt to assess the effect of kahweol on triple-negative breast cancer cells, as well as the underlying mechanism and the implication of CTGF in this effect.

The authors have conducted a large number of experiments, but my main concern is that some of the data presented have been already reported in previously published studies.

1. The implication of CTGF in the regulation of proliferation and migration of triple-negative breast cancer cells is known and has been reported previously in a study not cited by the authors [Oncogene volume 40, pages 2667–2681 (2021)]. The particular study should be cited here.  

2. The anti-tumour effect of kahweol on breast cancer cells has been shown.

3. I feel that the authors should isolate among their findings their novel results, focus on them (i.e., the association between 1 and 2, that is the mode of action of kahweol against breast cancer that passes through CTGF signaling) and re-organize the manuscript.

4. There are some grammar and syntax errors, some missing spaces before references and some inconsistencies regarding font and bibliography presentation formatting that need to be corrected (for examples, please refer to the uploaded file).

5. Ln93 and throughout: Please rearrange Figures’ order according to their order of appearance in the text. In this regard, this should be supplementary Figure 1.

6. Figure 1B is not readable. Please magnify at least the part of interest, otherwise, Figure 1B may be deleted.

7. Authors quote various RNA-seq data without any validation of selected DEGs with RT-qPCR analysis.

8. Ln130: Please elaborate. Do the authors mean "next" instead of "below"?

9. In supplementary Table 1 it would be more informative if authors presented fold change instead of log fold change.

10. Ln237-238: Please refer to the respective Figure or Table.

11. In Figure 5 and Supplementary Figure 1, please add units in y-axis.

12. Why is not a mean ± SD is not presented in the densitometric analysis of Figure 5B?

13. Ln282-283: Please add a reference.

14. Ln310: This has been already previously reported.

15. Ln361: No experiments and results are presented here using SKBR3 cells. Please check.

16. Ln477: Please elaborate. Were the cells pre-treated with kahweol before being exposed to CTGF? If yes, how long did pre-treatment last? Were cells depleted from kahweol before CTGF addition in the cultures or did kahweol remain for the entire 48-h incubation?

Comments on the Quality of English Language

There are some grammar and syntax errors, some missing spaces before references and some inconsistencies regarding font and bibliography presentation formatting that need to be corrected.

Author Response

(The authors gave the same response as above.)

Reviewer 4 Report

Comments and Suggestions for Authors

Unraveling Connective Tissue Growth Factor as a Therapeutic 2 Target and Assessing Kahweol as a Potential Drug Candidate in Triple Negative Breast Cancer Treatment

The present study demonstrates CTGF as a potential prognostic marker for guiding TNBC treatment and suggests kahweol as a promising antitumor compound capable of regulating CTGF expression to suppress cell motility in TNBC. The approach and the overall design of the study are good. However, the authors should address the following concerns.

1.      Materials and methods section needs to be revised with proper details of drug treatment, and duration of the treatment.  Also, how did the authors fix the dose of kahweol?

2.      What is the vehicle used for the treatment of kahweol? Add the details about the solubility of kahweol in drug preparation section.

3.      Should include the details about the bioavailability of kahweol.

4.      Did the authors use the same membrane to develop the protein of interest and housekeeping gene? Also, for phosphoproteins and corresponding proteins? The original blot images look like those results are not from the same membrane!!!

5.      Authors have shown random lanes for the respective treatment groups. Eg: p-P38 & P38. Not only the membranes are different, but also the order of lanes is random. This raises a question about the authenticity of the data. This should be clearly addressed with proper blots. Then only the study will be authentic and the findings and conclusion can be valid.

6.      Add limitations of the study.

Author Response

(The authors gave the same response as above.)

Round 2

Reviewer 1 Report

Comments and Suggestions for Authors

The authors clearly and completely replied to all my concerns. The manuscript is now suitable for publication on IJMS.

Reviewer 2 Report

Comments and Suggestions for Authors

The authors have made changes or provided compelling arguments. This reviewer is satisfied.

Reviewer 3 Report

Comments and Suggestions for Authors

This is the revised version of a previously submitted manuscript. The authors have addressed most of my concerns raised during the previous round of the reviewing process.

Comments on the Quality of English Language

Quality of English has much improved in comparison to the original version.

Reviewer 4 Report

Comments and Suggestions for Authors

Eve though the explanation for using different membranes for the detection of phosphorylated and corresponding total protein is not satisfactory, authors have addressed all other questions the revised the manuscript accordingly.